# A Preliminary Evaluation of Mobile Phone Apps to Curb Alcohol Consumption

**DOI:** 10.3390/ijerph19010135

**Published:** 2021-12-23

**Authors:** Omar Mubin, Billy Cai, Abdullah Al Mahmud, Isha Kharub, Michael Lwin, Aila Khan

**Affiliations:** 1School of Computer, Data and Mathematical Sciences, Western Sydney University, Penrith 2751, Australia; o.mubin@westernsydney.edu.au (O.M.); billycai@live.com.au (B.C.); 2Centre for Design Innovation, Swinburne University of Technology, Melbourne 3130, Australia; aalmahmud@swin.edu.au; 3School of Business, Western Sydney University, Penrith 2751, Australia; I.Kharub@westernsydney.edu.au (I.K.); M.Lwin@westernsydney.edu.au (M.L.)

**Keywords:** mobile app, user needs, persuasive design, Mobile App Rating Scale (MARS), alcohol cessation

## Abstract

Mobile apps have become increasingly prevalent in modern society, and persuasive technology has a broader market than ever. Mobile-based alcohol cessation apps can promote positive behaviour change in users and improve the overall health of our society. This research aimed to understand the various features users respond to and make design recommendations for alcohol cessation apps. This paper reports on three sources of feedback (user ratings, user reviews, MARS App Quality score) provided on 20 alcohol cessation apps in the Google Play Store. Our findings suggest that self-control type apps received much greater positive user reviews than motivational apps. In addition, this trend was not observed through numeric user ratings. We also speculate on design recommendations for apps that are meant to inhibit alcohol intake.

## 1. Introduction

Alcohol consumption is amongst the five leading risk factors that are responsible for one-quarter of deaths in the world and one-fifth of disability-adjusted life years [1]. Alcohol consumption led to 288 million deaths per year and was the leading risk factor for premature death and disability among people aged 15–49 years [2]. It leads to far more health loss for males than for females. Further, there is a strong correlation between alcohol use and the risk of cancer, injuries and communicable disease. Therefore, there is a call to revisit alcohol control policies and health programs and consider recommendations for abstention [3].

Excessive use of alcohol is a public health concern, and its foundations in our society are widespread and visible throughout many age groups. Alcohol consumption has negative consequences not only for the individual but also for friends or family and the workplace. Antai, Lopez [4] found men were more likely to report alcohol-related harm than female drinkers when alcohol consumption was measured either as a frequency or as drinks per occasion. Additionally, there is a positive association between binge drinking and poor physical and mental health. Though society generally tolerates its use, many issues can arise, such as binge drinking, drink driving, unsafe sex and violence [5].

Excess alcohol consumption leads to complications such as alcohol withdrawal, and many find it challenging to cope with drinking. The symptoms of severe alcohol consumption usually manifest as a loss of control and physical dependence. In the past few decades, alcoholism treatment is being thoroughly investigated and made accessible [6].

Following the release of app stores on mobile devices in 2008, smartphone applications have become an increasingly vital part of modern society. These apps cover a broad range of categories such as games, lifestyle and finance and have become tools that a large proportion of the working society regularly relies on. The influences of mobile apps have been steadily increasing since their inception, and it has become a mainstream way of spreading and gathering information. This is owed significantly in effect to the undeniably significant impact that social networking has had on introducing people from all age ranges to regularly using current technology.

An emerging category of mobile apps aims to change user attitudes and behaviours through persuasion and social influence rather than coercion and deception. These apps are categorised as persuasive technology and utilise particular features and principles in order to induce positive behaviour change in users [7]. Such apps work in two main ways: (a) self-monitoring of user behaviour and (b) providing social support [8]. Self-monitoring refers to the regular logging of consumption-related information, such as the amount consumed, time of consumption and expenditures. Many people with problematic alcohol consumption never seek help [9]. Self-help tools have been shown to work effectively in a number of cases [10].

Cunningham and Blomqvist [11] found that treatment services are only used by a minority of people with alcohol dependence, but those who do access them tend to use a range of different alcohol treatment services in the same year. The major reason for not seeking treatment was a lack of acknowledgement of the problem. Other reasons include stigmatisation [1], patients not believing that the treatment will be effective or not knowing about the treatment options [12]. Additionally, other reasons for not seeking treatment were that seeking treatment would require total abstinence, treatment with Disulfiram or residential treatment. These options were considered unappealing and made this lack of knowledge a barrier to treatment seeking. It is essential that awareness regarding treatment options is improved in order to lower the barriers to treatment [13].

In view of the findings from previous studies, record-keeping in a digital diary format may be seen as an effective tool for users to monitor their drinking behaviour. Conversely, perceived social support is associated with overall lower use of alcohol [14]. Apps such as [15] serve as a platform for like-minded people to share experiences and encourage each other in dealing with drinking problems.

Persuasive technologies such as mobile phone apps have the edge over humans in terms of their persuasive power. A major advantage of such technologies is the extent of anonymity assured to users [16]. Moreover, technologies are ‘scalable’ and capable of being easily upgraded [17]. A mobile phone app—unlike a human service provider—can handle a large volume of users simultaneously.

The usefulness of app reviews can be gauged from two perspectives. First, app reviews can be helpful for other users, and positive reviews about an app may translate into greater usage by new customers or users. It has been reported that online reviews by customers are ranked to be as trustworthy as personal recommendations. Second, reviews are essential for app developers, as any new design or features can consider user requirements as reflected in their reviews [18].

In this study, we attempt to analyse current alcohol cessation apps on the Android platform based on their user reviews, in-store ratings and a researcher rating by using the MARS scale [19] for rating mobile apps.

MARS has been used widely to evaluate the quality of health-related mobile apps [20,21]. MARS has three sections: app classification, app quality ratings and subjective app quality. The app classification section gathers descriptive and technical information about the app. The app quality ratings section has four dimensions (engagement, functionality, aesthetics and information), and each of them is rated on a 5-point scale (1 = inadequate to 5 = excellent). The overall app quality is presented as the mean score of the four dimensions.

We also attempt to determine which strategies and features users found most useful and suggest features that further apps in this field should consider. The article is organised as follows: first, there is a review of the available apps; next, there is a discussion on app features and designs, which is then followed by the methodology followed and findings of this study. The paper concludes with possible research directions in future.

### 1.1. Current Apps in the Market

With the ubiquity of the internet and the proliferation of smart devices, different apps are available to help combat alcohol use disorder. Earlier in the development phase, health interventions were limited to making phone calls and using short message services (SMS). There has now been a significant shift towards using other functions and features. Apps can now deliver user-friendly information by tapping into a plethora of online resources. It is also feasible to set personalised reminders as well as use self-assessment tools for goal setting and tracking. Most of all, geolocation services can now identify and flag ‘high risk’ locations to users with their consent [22].

A recent study of alcohol consumption apps [23] grouped them into two categories: drink driving prevention and alcohol management. This study revealed that while drink driving prevention apps were downloaded 3.5 times more than the other type, they were significantly less engaging. More importantly, it was found that some of the apps using calculators for blood alcohol content (BAC) before driving were inaccurate.

A particular study investigated the impact of mobile interventions on the drinking habits of university students [24]. While it concluded that the apps could provide accessibility to a large number of students, the apps used did not seem to influence alcohol consumption. However, previous studies using the internet have shown that web-based interventions were successful in lowering alcohol dependency [25]. This implies that although the potential for mobile apps to be successful in this field exists, it remains undetermined which features are necessary.

It is important to consider the current state of these alcohol cessation apps in the market before deciding on the key features. One of the earliest studies in this field investigated alcohol-related apps available on the iTunes store [26] and found that very few apps addressed the topic of positive behavioural change in the user. This sentiment has been carried on through many other studies. Similar research was carried out by David Crane [27], who found that few alcohol-related apps attempted to reduce alcohol consumption, and the majority either implicitly or explicitly endorsed its use. These findings reinforced the conclusion of previous research [28], which also identified that the majority of popular apps in this category encouraged the consumption of alcohol and the blood alcohol concentration (BAC) measuring apps tended to provide unreliable information. Of the 384 apps included in their study, only 11% (44) were explicitly intended to promote healthy alcohol drinking habits among users. These studies show that the possibilities and impact of mobile apps on helping individuals overcome alcohol drinking habits still largely remain unidentified and that alcohol-related apps, in general, are quite neglected in the current app market. This is possibly due partly to limited awareness that these apps actually exist, and as a result, people who seek help would not consider searching for an app that can provide this. This is in stark contrast to exercise apps, which most smartphone owners are aware of and actually use [29].

### 1.2. App Features and Design Considerations

It is necessary to ascertain which features an effective app should have. Firstly, guidelines already exist for the treatment of alcohol problems [30]; however, it is unknown if apps follow these guidelines. Related studies have provided content analysis in apps for smoking cessation and discovered that these apps rarely adhere to the guidelines for smoking [31]. It is vital to stick to these guidelines, as they are the culmination of many years of testing, a necessity for this field that is still in its early years with limited research behind it. Furthermore, completely ignoring these guidelines can lead to a myriad of problems. One such example we must consider when designing these apps is the importance of providing correct information. While making false claims, such as, ‘Drinking more than one glass of alcohol a day will lead to cancer and reduce lifespan by 20 years’, could yield a decrease in alcohol consumption of users, this would definitely be interpreted as deceiving the users and undermining the principles of persuasive technology [17]. Rather, apps should provide strong evidence from research to persuade users [32]. In addition to this, there are many ethical concerns and complications that would arise if an app did use deception under the guise of educating the user.

There have been several studies focused on discovering which features appeal to users. Studies have shown that users showed better progress in apps that were more engaging [33]. From this, it can be inferred that users show greater progress in apps that are more appealing to them. It is therefore vital to discern which features users prefer in order to produce better apps. However, despite user reviews being such an important aspect, aside from conducting surveys and studies on specific apps, the majority of the feedback can only be gathered through app ratings and comments. Additionally, it is difficult to gather proper insights into these reviews, which is a necessity due to the likelihood of spam and useless comments provided on these apps [34].

Since apps targeting alcohol drinking habits have only started emerging recently, it is important to understand the current influence these apps have on society. Similarly, no real framework has been developed that defines the features that these apps should have. These apps are essential because they provide an additional treatment platform for patients that do not respond to traditional alcohol treatments, such as medications or Alcoholics Anonymous groups.

## 2. Materials and Methods

The current study inherits its methodology from prior work [35]. In the mentioned study, it was found that apps, which employed self-control, were more preferred amongst users due to their user interface (UI) and design features in comparison to apps, which employed a motivational strategy. However, that study focused solely on apps in the iTunes store, which meant that the user base was limited to people who used Apple products and only relied on user reviews to conclude. Given the popularity of the Android platform [36], it is imperative to closely investigate the user experience across Android-based alcohol cessation apps. Furthermore, in contrast to the previous study [35], we do not rely only on user reviews but aim to triangulate our results across two other sources of data, namely, user ratings and the self-report/personal MARS scale. This rating was carried out by using the Mobile App Rating Scale (MARS), a tool that was developed recently due to the lack of current resources to properly rate app quality on a global standard [19]. In summary, we aim to establish different ways of deducing app quality. The host institution provided an exemption from formal ethical approval due to the nature of the research and data inquiry. The overall process of data collection and analysis is illustrated in Figure 1.

### 2.1. Phase 1: App Selection

An exhaustive list of apps was gathered by entering alcohol-reduction-related keywords into Google Play. The search keywords used were: quit alcohol, alcohol recovery, alcohol self-help, sober app, substance recovery, quit drinking and stop drinking in order to obtain apps. The apps matching the inclusion criteria were recorded in an Excel spreadsheet as they were found along with the keyword used to find them. Any apps that were already listed on the spreadsheet were not included even when they showed up when searching later keywords. Further keywords were also used; however, they did not produce any exclusive results. It is possible that some combination of keywords may have produced additional apps that were not discovered in this study; however, it is believed that the list gathered was quite exhaustive in regards to our specific criteria.

The apps gathered using these keywords were filtered with certain exclusion criteria before they were recorded. These criteria required that the apps: be mostly written in English (especially the user interface-UI), be free to download apps, intend to reduce user alcohol consumption by promoting positive behaviour change and have a minimum of at least one review. This meant that apps that were only advertised to measure blood alcohol concentration (BAC) were not included because they did not explicitly intend to induce positive behaviour change in users. Our search resulted in a total of 20 apps (Table 1). At this juncture, we would also like to state that our list of apps was uniquely different from the iTunes apps shortlisted in [35]. We reached this conclusion by comparing the names of the apps across the list in [35] and our own list.

### 2.2. Phase 2: App Quality Rating

In the next stage of our study, we focused on the ratings of each application. A protocol was developed and strictly followed to provide personal ratings for each application. The steps followed in the protocol were: (1) download and install the application from the Google app store, (2) use the application for 15 min to understand the UI and how the app intends to reduce user alcohol consumption, (3) fill in the Mobile Application Rating Scale to produce an app quality mean score along with an app subjective quality score and enter these ratings into the Excel spreadsheet (main rater BC) and (4) step 3 was repeated by another researcher (OM) for five (25%) of these apps.

### 2.3. Phase 3: User Comments

This phase required analysing the comments and reviews of the apps left by the users on the app store. The apps were categorised based on the general strategy, i.e., self-control or motivational apps [35]. ‘Self-control’ apps relied on a strategy in which the user would be motivated by the progress of their own results (displayed on the app through tracking days or certain milestones reached) while ‘motivational’ apps gave the user motivation from sources within the app (such as daily messages, other users or education). Since there was a decent amount of crossover (such as tracking apps also providing features such as daily messages), the apps were sorted based on the primary strategy that they employed. The comments for all apps were extracted from Google Play Store servers by using a parsing script. In addition, the number of ratings for the app and the average rating (out of 5) was also recorded. The scripts to extract comments and rating observations were executed on a single day to avoid discrepancies in the data. An attempt was made to code and analyse all comments associated with each app; however, an upper limit of 100 random comments was set if the number of comments exceeded 100. From the human–computer interaction literature [37], we found examples of reducing the available dataset (such as message board posts) to manageable proportions if it is considerably large originally.

Similar to [35], the comments were then analysed based on the feedback towards the strategy and the UI/features (the execution of the strategy) of the app as follows:Comments that gave positive feedback regarding the strategy or the idea of the app were given a positive valence of +1, comments that gave negative feedback were given a negative valence of −1 and comments that were irrelevant or neutral were given a valence of 0.Comments that gave positive feedback concerning the UI and features of the app were given a positive valence of +1; comments that gave negative feedback, suggested improvements or stated problems with the app itself, such as crashing or bugs were given a negative valence of −1; and comments that were irrelevant or neutral were given a valence of 0.We also qualitatively analysed the content of the comments to determine the most useful or desired features of the applications that we later summarised in the discussion section.

A series of non-parametric and significance testing analyses were outlined. These included Chi-squared tests to assess the impact of app type on public user ratings and determine the association between app type and valence of feedback on user interface and behavioural strategy. An analysis of variance (ANOVA) was used to determine the effect of app type on the subsequent two-prong MARS ratings.

## 3. Results

As mentioned earlier, 20 free apps from the Google Play Store were shortlisted and identified. We did not conduct any analysis on paid apps, and this should hence be considered a limitation of our analysis. Each app was classified as motivational or of self-control type, and this functioned as the primary independent variable of our analysis. Nine apps were classified as of motivational type.

*User Ratings.* We treated the average user rating as ordinal data and conducted a Chi-squared test. However, the type of app did not have a significant association with the average rating (*p* = 0.63). An ANOVA was run to investigate if the total number of ratings varied across the types of apps. However, this was also not significant (*p* = 0.46). Therefore, one can speculate that users were equally active and involved with reviewing both types of apps.

*MARS App Quality Rating.* We analysed the reports from the MARS App Quality Survey to deduce if the type of app influenced how apps were being evaluated by experts (or, in this case, the authors of the paper). We computed the standard deviation for the difference across ratings for 25% of the apps (five in number) as evaluated by two raters. The standard deviation for the difference across the two raters for app mean quality was 0.27, and the standard deviation for the difference across the two raters for app subjective quality was 0.99. We then proceeded with utilising the ratings of Rater 1 for subsequent ANOVA analysis. The type of app did not influence either scale (app mean quality—*p* = 0.64 and app subjective quality—*p* = 0.36). We did notice that self-control apps were performing marginally better, but of course, this was not significant. Table 2 summarises the ratings:

*Comments Analysis.* In total, 884 comments across the 20 apps were analysed. The average length of a comment was ~34 words. The maximum length of a comment was 133 words, whereas the minimum was one word. Two researchers independently coded 20% of the sample. In order to verify our coding scheme, Cohen’s kappa was computed for their codes. The kappa for valence on behavioural strategy was 0.68, and the kappa for valence on the user interface was 0.63. This indicated moderate agreement, and discrepancies were resolved through agreement. Thereafter, one of the researchers finalised all the other codes (Table 3). Two Chi-squared tests were run separately (one each on the two valence measurements) on the completed codes. The type of app had a significant association with the behavioural strategy valence measurement (*p* = 0.001), such that self-control apps received significantly more positive comments on their ability to instigate behavioural change as compared with motivational apps (residual z = 3.1). In addition, the type of app had a significant association with the user interface valence measurement (*p* = 0.002). Self-control apps received significantly more positive comments on the effectiveness of their user interface than motivational apps (residual z = 2.8).

## 4. Discussion

In this section, we discuss the implications of our results and speculate on the primary reasons for their occurrence. Quotes from user reviews are provided in brackets and double quotes. From Table 3, it can be observed that self-control apps received significantly more positive comments as compared with motivational apps. This suggests that either the overall quality of self-control apps in the market was higher or the behavioural strategy requires fewer features in the app to promote positive behaviour change within the user. Thus, it is possible that while users do not mind which strategy these apps decide to use, the apps that fall into a motivational category require more features or functionalities to satisfy the user. This is understandable because self-motivated apps rely mainly on users viewing their own progress and, therefore, usually require only a tracker to function (‘Great aid for tracking progress’ and ‘I like knowing exactly how much progress I’ve made instead of having to think of a ballpark estimate’). Furthermore, the number of positive feedback for both categories in UI/features was much lower than the positive feedback for the behavioural strategy. Similarly, the negative feedback was much higher on the UI/features compared with the behavioural strategy. This could mean that users believed that the strategy of these apps was acceptable in theory or as a concept, but the apps still required a great deal of improvement as far as their user interface was concerned. Although implementing self-control apps would probably be easier, the high negative review count on self-control apps leads us to believe that the behavioural strategy was preferred by the users. It is still too early to draw any conclusions about these apps; however, early results suggest that users are generally more satisfied with current apps that employ a self-control strategy, possibly because these apps usually only consist of a function to track sobriety days and users like to see this visible progress. Apparently, users do not find this lacking but simplistic and easy to use for their own goals.

One of the key goals of our study was to establish different mechanisms of determining app quality or app perception. Whilst we found significant associations through text-based user reviews, we did not find any significant trends relating to user ratings or researcher-based MARS ratings. This apparent weak correlation of data leads us to speculate that deep analysis of user-based qualitative review data on user apps may be required rather than simple rating or ticking mechanisms.

In a review of iTunes alcohol cessation apps, an analysis of the comments showed users preferred the UI and design features of apps that employed a self-control strategy compared with a motivational one [35]. These findings are consistent with other studies that found promoting self-monitoring to lead to better outcomes [38,39], and similar results are shown for long-term weight management [7]. These studies show that self-monitoring and control are probably key motivators in long-term positive behaviour change. These apps must remain relevant to the user and prevent relapse of habits while providing treatment. Therefore, with the evidence that this literature has provided, it is recommended that future apps in this field be designed around a principle that relies on self-monitoring and control.

We would also like to contemplate suggestions of additional features that future apps in this field should consider based on a meta-analysis of the user reviews. Such features are tutorials (because many comments expressed their inability to use the app properly), offline functionality (users liked the convenience), easy-to-read text and no advertisements. Some app users stated that certain apps had advertisements for alcohol, which distracted them severely from achieving their goal of limiting consumption (‘Still showing intermittent alcohol ads’). However, an overwhelming opinion seemed to support the tracking function of these apps, which is supported by the high ratings of apps that essentially consist of only a calendar or a counter/tracker for sobriety days (‘Love it!—The counter is a great tool for me to see how far I’ve come’ and ‘Only been using it for a week but very satisfying to mark off another sober day and watch the month slowly go green’).

Functionality is possibly the most important aspect when creating alcohol cessation apps. Any app that has bugs or crashes will not help users and is likely to frustrate them instead. Thus, before considering adding any features to apps, the app must be stable and function properly on all devices. Several users stated the importance of portability (particularly relevant for the Android platform). On several occasions, the users reported the app malfunctioning when ported to another mobile device (‘Not working for me on Galaxy S3, I’m having issues with this app’ and ‘Crashes on Nexus 5X’).

### Limitations

Although the study attempted to minimise bias and other potential problems, it is possible that some areas in the study produced a biased result due to the subjective nature of rating using the MARS scale. Furthermore, the responses allowed are also quite limited, with the possible responses to the latter question being ‘1—No, 3—Maybe, 5—Yes’. The effect of both limitations combined are likely to cause extreme fluctuation of marks between users and between the app quality and app subjective quality mean score on the same application, making the subjective quality score much less useful than it would seem. Another possible limiting factor is the particular question ‘Credibility: Does the app come from a legitimate source?’, with higher scores being awarded to government or university sources. Although somewhat minor, the source of the app itself should not be a contributing factor to whether the app is rated better or worse in effectiveness for achieving the user’s goals. The MARS also favours more complex apps than apps that are simple and have fewer features but present their content more succinctly. Similarly, the numeric star-based rating mechanism could have been viewed vaguely by some users—we attempted to overcome this bias by treating it as an ordinal variable.

Analysis of the comments also had a few limiting factors. Some of the comments saying (‘Nice app’) or similar were difficult to analyse because it was difficult to tell whether they were praising the strategy the app utilised or the features of the app itself. In most cases, they were assigned as a positive valence for both strategy and UI/features. If an app was praised but improvements were also suggested, apps were also assigned a negative valence because this meant that the user was not completely satisfied. This mostly occurred in the UI/features category in which users liked one aspect but thought the app could improve, such as by being able to track another habit at the same time (‘Good ideas in this app, An Improvement would be to allow the editing of notes’ and ‘Like it but it could improve the separate settings for the widget’). Furthermore, apps were usually praised for being ‘simple’ and ‘easy to understand’. Therefore, it was difficult to exactly discern which features were and were not required. Finally, not all of the users were using the apps to help overcome their drinking habits. This was most prevalent in apps that tracked habits in general. Due to many comments not specifying which habit they were regarding, it was impossible to discern from the limited message being conveyed by the comments. Lastly, we also acknowledge we have not considered iTunes data, as this could have served as an interesting comparison point. However, we have already reported on iTunes apps individually elsewhere.

## 5. Conclusions

In this paper, we investigated which aspects users respond positively or negatively to and suggested features that should be considered when creating alcohol cessation apps. Our preliminary findings suggest that self-control type apps received much more positive user reviews than the motivational type of apps. In addition, this trend was not observed through numeric user ratings on both types of apps. We have also found several key features for alcohol cessation apps such as tutorials, offline functionality, easy-to-read text and no advertisements. As future work, we aim to directly compare our results with a similar detailed analysis for alcohol cessation-based iTunes apps. In addition, we aim to compare alcohol cessation apps with other motivational apps such as weight loss apps and deduce the common and different features.

## Figures and Tables

**Figure 1 ijerph-19-00135-f001:**
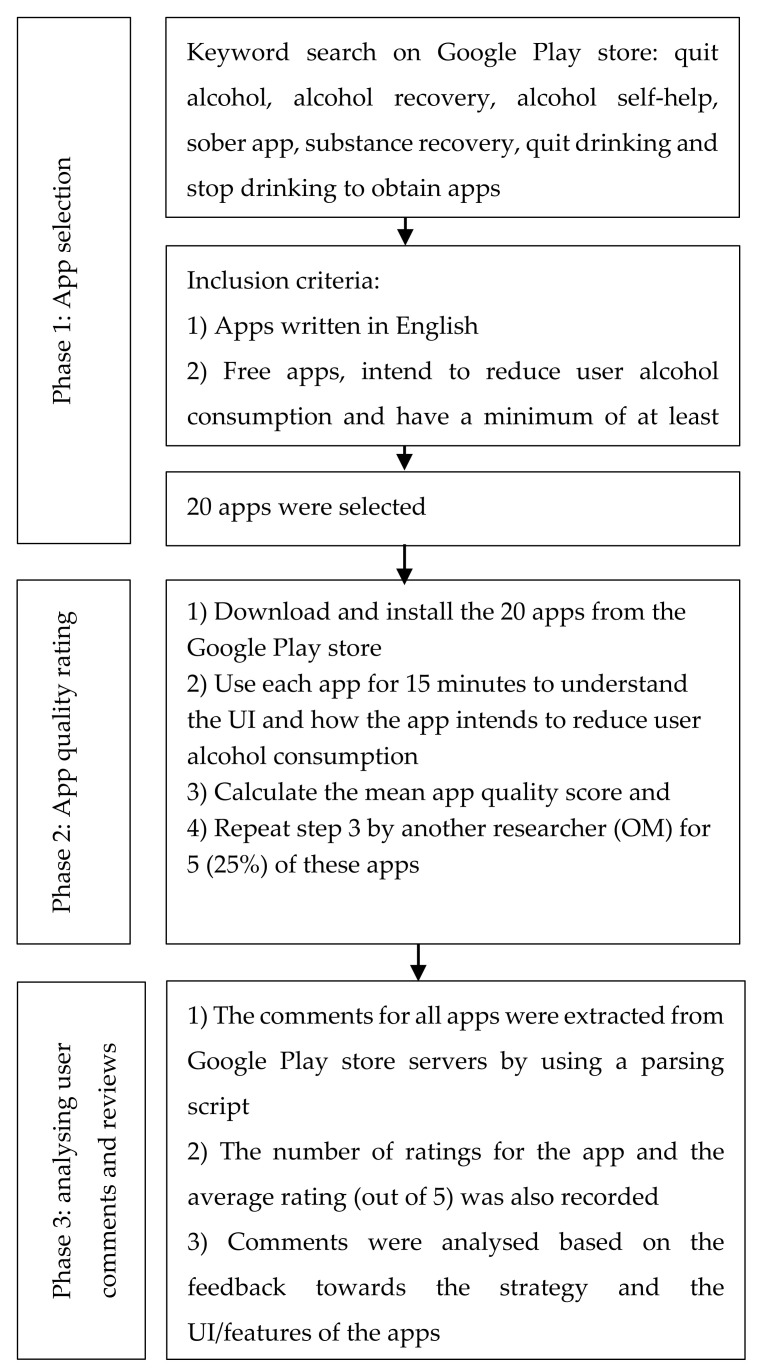
Flow diagram of the data gathering and analysis process.

**Table 1 ijerph-19-00135-t001:** Table summarising the 20 apps.

Self-Control Apps	Motivational Apps
Sober Time Stop Drinking Alcohol App Sober Tool Sober Time Counter HabitBull CleanTime Counter Quit Addiction Sobriety Counter AlcoChange Sober Today Addiction Free IMQuit	Abolish Alcohol Hypnosis Rehapp Self Help Daily Reflections Sobriety Clock Sober Grid Recovery Today Sober Day Clean and Sober Time

**Table 2 ijerph-19-00135-t002:** Table summarising MARS ratings.

App Name	Rater 1 (BC)		Rater 2 (OM)	
	App Quality Mean Score	App Subjective Quality Score	App Quality Mean Score	App Subjective Quality Score
Sober Time—Sobriety Counter	4.43	3.75		
Stop Drinking Alcohol App	3	1	2.2	1
SoberTool Sober Time Counter	4.02	3		
HabitBull—Habit Tracker	4.88	4.75		
CleanTime Counter	3.11	1.75	2.95	1.5
Abolish Alcohol Hyponsis	3.31	2.25	3.123	3
Rehapp	4.33	2.75		
Quit Addiction: iQuit App	3.8	2.25		
Clean and Sober Time	3.82	2.25		
IMQuit Addiction	4.15	2.75		
Self Help Just for Today	3.4	2		
Daily Reflections	2.66	1		
Sobriety Clock	3.3	1	3.15	3
Sobriety Counter	2.94	1.25		
Alcochange—Alcohol Tracker	4.20	3.25		
Sober Today	3.34	2.25	2.75	2.75
Recovery Today	2.88	1.75		
Sober Day Recovery App	3.02	1.25		
My Sober Life	4	3.75		
Addiction Free	3.67	2.5		

**Table 3 ijerph-19-00135-t003:** Table summarising the category and feedback towards the 20 apps.

		Valence of Feedback Towards Employed Behavioural Strategy			Valence of Feedback Towards App UI/Features		
		Positive	Negative	Irrelevant	Positive	Negative	Irrelevant
Category of App	Self-control	219	58	198	80	112	283
	Motivational	144	47	218	47	80	282

## Data Availability

Not applicable.

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
