# Peer review of "A Preliminary Evaluation of Mobile Phone Apps to Curb Alcohol Consumption"

_ijerph, 2021, doi:10.3390/ijerph19010135_

Round 1
Reviewer 1 Report
The article by Mubin et al addresses an interesting issue related to the use of apps for alcohol consumption cessation. Overall, the paper is well structured, and given the latest data published by the WHO regarding alcohol consumption, I consider it current enough.
I think the article is worthy of publication once these minor revisions are made.
INTRODUCTION
Line 22-30
The effects and risks induced by alcohol intake, as well as the different patterns of alcohol consumption, should be reviewed considering recent literature.
doi: 10.1155/2014/853410
https://doi.org/10.2471/blt.09.070565.
https://doi.org/10.1016/s0140-6736(18)31310-2
Line 47
Authors might consider including more recent bibliographic notes. https://doi.org/10.1093/alcalc/agl081.
https://doi.org/10.1186/s13011-015-0028-z.
https://doi. org/10.3109/10826084.2014.891616.
Line 124
The claim made (Drinking more than one glass of alcohol a day will lead to cancer and reduce lifespan by 20 years) must be supported by the literature.
https://www.aicr.org/cancer-prevention/food-facts/alcohol/
Line 155
You should give definition of MARS (Mobile App Rating Scale) when you mention for the first time in line 66
Line 173
Please define UI when you mention it for the first time
MATERIALS AND METHODS
Please add a flow diagram of the study protocol
Authors could include a "Statistical Analysis" paragraph before the results, in which they list the types of statistical analysis performed.
RESULTS
The author might consider schematizing the results obtained.
Author Response
Reviewer 1:
The article by Mubin et al addresses an interesting issue related to the use of apps for alcohol consumption cessation. Overall, the paper is well structured, and given the latest data published by the WHO regarding alcohol consumption, I consider it current enough.
I think the article is worthy of publication once these minor revisions are made.
Our response: Many thanks for the reviewer who appreciated our approach and provided helpful comments.
INTRODUCTION
Line 22-30
The effects and risks induced by alcohol intake, as well as the different patterns of alcohol consumption, should be reviewed considering recent literature.
doi: 10.1155/2014/853410
https://doi.org/10.2471/blt.09.070565.
https://doi.org/10.1016/s0140-6736(18)31310-2
Our response: We have included the suggested references in our revised manuscript.
Please see introduction section of the manuscript (line 22-30)
Line 47
Authors might consider including more recent bibliographic notes. https://doi.org/10.1093/alcalc/agl081.
https://doi.org/10.1186/s13011-015-0028-z.
https://doi. org/10.3109/10826084.2014.891616.
Our response: We have summarised the recommended literature. Please see the introduction section of the manuscript (line 32-39)
Line 124
The claim made (Drinking more than one glass of alcohol a day will lead to cancer and reduce lifespan by 20 years) must be supported by the literature.
https://www.aicr.org/cancer-prevention/food-facts/alcohol/
Our response: We have added the supporting statement. Please see line162 to 166.
Line 155
You should give definition of MARS (Mobile App Rating Scale) when you mention for the first time in line 66
Our response: We have defined MARS. Please see line 97 to 102.
Line 173
Please define UI when you mention it for the first time
Our response: Thanks for the suggestion. We have defined the acronym UI in the manuscript. Please see line 187 and 259
MATERIALS AND METHODS
Please add a flow diagram of the study protocol
Our response: we have added a flow diagram. Please see Figure 1.
Authors could include a "Statistical Analysis" paragraph before the results, in which they list the types of statistical analysis performed.
Our response: As suggested, we have included the following paragraph in the manuscript (see line 312-316)
A series of non-parametric and significance testing analysis were outlined. These included Chi-Square tests to assess the impact of app type on public user ratings and determine the association between app type and valence of feedback on user interface and behavioural strategy. An Analysis of Variance (ANOVA) was used to determine the effect of app type on the subsequent two-prong MARS ratings.
RESULTS
The author might consider schematizing the results obtained.
Our response: We have revised the results section and added a new table (Table 2) to clarify the results around the MARS scores.
Reviewer 2 Report
The manuscript presents a preliminary evaluation for mobile phone apps regarding alcohol consumption. The paper is interesting and it describes the methodology adequately. However, the quality of the paper should be improved in parts for publication. 1. Authors should consider to modify the title for a better reflection to the manuscript content. The title is likely misleading readers to that this manuscript is about to discuss how apps curb alcohol consumption and to evaluate the efficiency of those apps on curbing alcohol consumption. The fact is however the manuscript only focused on the rating and review of apps from users. Given the manuscript content, the appropriate title could be “Evaluation of Mobile Phone Apps Regarding Alcohol Consumption……”. 2. It is a major drawback that the study did not include iphone users, leading an incompleteness of survey. 3. Authors should improve the description on filling in the Mobile Application Rating Scale to produce an app quality mean score. For example, the rating scale is based on what features of an app and the following calculation. Also, authors may consider to display the distribution of the resulting scores. 4. The “raters” mentioned on page6 and how the rater perform evaluation are unclear.Author Response
Reviewer 2:
The manuscript presents a preliminary evaluation for mobile phone apps regarding alcohol consumption. The paper is interesting and it describes the methodology adequately. However, the quality of the paper should be improved in parts for publication.
Our response: Many thanks for the reviewer who appreciated our approach and provided helpful comments.
- Authors should consider to modify the title for a better reflection to the manuscript content. The title is likely misleading readers to that this manuscript is about to discuss how apps curb alcohol consumption and to evaluate the efficiency of those apps on curbing alcohol consumption. The fact is however the manuscript only focused on the rating and review of apps from users. Given the manuscript content, the appropriate title could be “Evaluation of Mobile Phone Apps Regarding Alcohol Consumption……”.
Our response: Thanks for your suggestion to modify the title. Our new title is “A Preliminary Evaluation of Mobile Phone Apps to Curb Alcohol Consumption”.
- It is a major drawback that the study did not include iphone users, leading an incompleteness of survey.
Our response: We have addressed this in the revised manuscript (see line 456-458).
Lastly, we also acknowledge we have not considered iTunes data as this could have served as an interesting comparison point. However, we have already reported on iTunes apps individually elsewhere
- Authors should improve the description on filling in the Mobile Application Rating Scale to produce an app quality mean score. For example, the rating scale is based on what features of an app and the following calculation. Also, authors may consider to display the distribution of the resulting scores.
Our response: We have added a paragraph to clarify the issue (see line number 97-103)
MARS has been used widely to evaluate the quality of health-related mobile apps (Knitza et al., 2019; Salazar et al., 2018). MARS has three sections: app classification, app quality ratings, and subjective app quality. The app classification section gathers descriptive and technical information about the app. The app quality ratings section has four dimensions (engagement, functionality, aesthetics and information), and each of them is rated on a 5-point scale (1 =inadequate to 5=excellent). The overall app quality is presented as the mean score of the four dimensions.
Table 2 has also been added to showcase the scores.
- The “raters” mentioned on page6 and how the rater perform evaluation are unclear.
Our response: we have clarified the issue in the revised manuscript (see line number 273 to 276). Also, see line number 97-103 to understand the rating scale. We have added a new table (Table 2) for clarifying the evaluation process by the raters. The first and second author acted as the raters of the apps. Initials of both have been added to the rating text.
Fill in the Mobile Application Rating Scale to produce an app quality mean score along with an app subjective quality score and enter these ratings into the Excel spreadsheet, 4) Step 3 was repeated by another researcher (OM) for 5 (25%) of these apps.
Reviewer 3 Report
This article is written by computer, design and business experts and it is not easy for a physician to rate the manuscript. The statistical method used to compare apps is correct. The results are reported very briefly. The discussion of the results appears coherent. From the doctor's point of view, it is interesting to know that self-monitoring and control are key motivators in long term positive behaviour change.
Author Response
Reviewer 3
This article is written by computer, design and business experts and it is not easy for a physician to rate the manuscript. The statistical method used to compare apps is correct. The results are reported very briefly. The discussion of the results appears coherent. From the doctor's point of view, it is interesting to know that self-monitoring and control are key motivators in long term positive behaviour change.
Our response: Many thanks for the reviewer who appreciated our approach and provided helpful comments.
we have added a new reference in the discussion section to emphasize how self-monitoring and control features are the key factors to change behaviour. Please see line 395 to 404.
Round 2
Reviewer 2 Report
My previous suggestions and questions have been addressed.